# Food Restriction in Anorexia Nervosa in the Light of Modern Learning Theory: A Narrative Review

**DOI:** 10.3390/bs13020096

**Published:** 2023-01-23

**Authors:** David Garcia-Burgos, Peter Wilhelm, Claus Vögele, Simone Munsch

**Affiliations:** 1Department of Psychology, University of Fribourg, 1700 Fribourg, Switzerland; 2Department of Psychobiology, The “Federico Olóriz” Institute of Neurosciences, Biomedical Research Centre, University of Granada, 18071 Granada, Spain; 3Institute for Health and Behaviour, Department of Behavioural and Cognitive Sciences, University of Luxembourg, 4365 Esch-sur-Alzette, Luxembourg

**Keywords:** anorexia nervosa, associative learning, conditioning, eating disorders, food restriction, modern learning theory

## Abstract

Improvements in the clinical management of anorexia nervosa (AN) are urgently needed. To do so, the search for innovative approaches continues at laboratory and clinical levels to translate new findings into more effective treatments. In this sense, modern learning theory provides a unifying framework that connects concepts, methodologies and data from preclinical and clinical research to inspire novel interventions in the field of psychopathology in general, and of disordered eating in particular. Indeed, learning is thought to be a crucial factor in the development/regulation of normal and pathological eating behaviour. Thus, the present review not only tries to provide a comprehensive overview of modern learning research in the field of AN, but also follows a transdiagnostic perspective to offer testable explanations for the origin and maintenance of pathological food rejection. This narrative review was informed by a systematic search of research papers in the electronic databases PsycInfo, Scopus and Web of Science following PRISMA methodology. By considering the number and type of associations (Pavlovian, goal-directed or habitual) and the affective nature of conditioning processes (appetitive versus aversive), this approach can explain many features of AN, including why some patients restrict food intake to the point of life-threatening starvation and others restrict calorie intake to lose weight and binge on a regular basis. Nonetheless, it is striking how little impact modern learning theory has had on the current AN research agenda and practice.

## 1. Introduction

Anorexia nervosa (AN) is one of the most common eating disorders (EDs). AN is a serious, often treatment-refractory mental illness, characterised by a distorted perception of body size and/or shape and the sustained attempt to restrict food intake (e.g., cutting back on the amount of food eaten, fasting or eliminating certain types of food) that leads to pathological weight loss [1]. Within the AN disorder, two subtypes have been defined: the restricting type (R-AN), which achieves weight loss by limiting caloric intake, and the binge–purging type (BP-AN), which presents both restriction and binge-eating and/or purging behaviour such as self-induced vomiting or misuse of laxatives (DSM-5, [2]). Lifetime prevalence for AN has been reported at 1.4% for women and 0.2% for men (see [3,4]), with a mortality rate of 5% within ten years [2,4,5,6].

Although substantial progress in the treatment of AN has been achieved, the efficacy of interventions is still limited. Five-year recovery rates are estimated at 69% and remission rates in randomised controlled trials range from 19% to 65% [4], with relapse rates of 9–52%, in line with an increase in time since treatment among those who achieved remission [7]. Therefore, considerable progress in the treatment of AN is still needed.

Learning theory is well placed to enable this progress. Learning models can explain the development and regulation of eating, as well as the psychological processes involved in the control over how much is eaten, in healthy people (cf. [8]) and in those with EDs [9]. Moreover, learning theory has been used as an interdisciplinary and translational platform that fosters the cross-fertilisation of ideas between basic and clinical research (cf. [10,11,12]). Learning theory provides convincing heuristics and testable models for mental disorders, which have demonstrated predictive and diagnostic validity [13]. Moreover, it offers a mechanism-oriented approach [14] that enables the integration of current findings from neuroscience and experimental psychopathology [15], which is a goal of recent initiatives for the new foundation of ED psychopathology (cf. the NIMH Research Domain Criteria [RDoC] [16]). However, although there has been a renewed interest in behavioural strategies rooted in the principles of learning to reduce ED symptoms [17], no comprehensive attempts have been made so far to determine the specific associative learning processes involved in the aetiology and maintenance of AN.

Consequently, this review provides an overview of what is currently known about the associative mechanisms underlying the sustained attempt to restrict food intake in AN, mainly focusing on negative emotional processes (disgust and fear). To do so, we first introduce basic concepts of learning theory and its historical progression concerning AN. Next, we describe what associative learning theory could mean for AN in terms of the associative analysis of pathological food rejection when conceptualised as avoidance behaviour, its potential to discriminate between AN subtypes and new avenues for intervention. Finally, we highlight to what extent current empirical findings are in line with this approach, as well as the questions that remain unanswered and how they should be investigated to move the ED field forward.

## 2. Method

This narrative review focused on prior work that has been central and pivotal to this specific topic and related to associative learning and AN, including empirical and conceptual papers. This was supplemented with a search strategy in the databases PsycInfo, Scopus and Web of Science (see Table 1 and Figure 1 for the strategy and term combinations for the search in databases and PRISMA 2009 flow diagram; see Appendix A for details). Inclusion criteria: peer-reviewed papers in English published until 2021, AN patients or animal models of EDs focused on AN and conditioning principles in animal and human research. Exclusion criteria: articles for which full text was not available, computational models and the neurobiological basis of AN, which were beyond the scope of this review. Finally, given that recent reviews on exposure therapy are available ([18,19], the terms “exposure” and “extinction” were not explicitly included in the search strategy. Additional references were identified in the articles retrieved in the first search round by performing a manual search.

## 3. Basic Concepts of Learning Theory

Learning theory is a coherent framework of integrated constructs and principles that describe, explain and predict how organisms learn and how this learning is translated into behaviour. Early in its history, learning theory was in harmony with the dominant behaviourist paradigm (cf. [20,21]), which only focused on what is directly observable, such as changes in external (motor or psychophysiological) responses. Behaviourist learning theory viewed learning as the result of experiences with two types of environmental relationships (see [12], for an introduction to learning and conditioning). One type of relationship occurs when two stimuli are experienced together, e.g., a tone is paired with an electric shock in a laboratory setting. This learning paradigm is called classical or Pavlovian conditioning. In this paradigm, the tone is initially a neutral stimulus as it does not produce the response of interest (fear). In contrast, the electric shock is the unconditioned stimulus that innately evokes an unconditioned response: fear. After the tone has been contingently followed by an electric shock several times (i.e., pairing of neutral stimulus and unconditioned stimulus), the tone will elicit fear even when no longer followed by the electric shock. At that moment, the tone has become a conditioned stimulus that evokes fear, which is then called the conditioned response. The other type of relationship occurs when an action is experienced followed by an outcome, and the related paradigm is known as operant or instrumental conditioning. An example of instrumental conditioning in a laboratory setting is jumping a barrier to avoid an otherwise imminent electric shock. The action (jumping a barrier) here is instrumental in avoiding the unpleasant outcome (the electric shock). It is important to note that most action–outcome relationships are only valid in the presence of a particular stimulus. This stimulus is called discriminative stimulus and becomes a signal that tells the organism what action is going to become reinforced. For example, a tone becomes a discriminative stimulus when it signals the availability of an electric shock if avoidance is not performed. Ultimately, the distinction between Pavlovian and instrumental conditioning is based on the type of events experienced and the experimental procedure used: conditioned stimulus → unconditioned stimulus versus discriminative stimulus → action → outcome.

Proponents of behaviourist learning theory intentionally ignored what goes on inside “the black box” of the learning organism [22]. Other authors opposed this approach, positing that the changes we observe in studies of learning may not directly mirror what the organism has learned. Thus, when the dominance of the behaviourist paradigm declined, internal processes gained recognition [23]. Today, modern learning theory explains changes in behaviour by internal processes during Pavlovian and instrumental conditioning in terms of associations between mental representations of stimuli and responses in memory (see Figure 2). Indeed, “conditioning is now described as the learning of relations among events so as to allow the organism to represent its environment” ([24], p. 151). With regard to mental representations, this term is used to refer to any model of external or internal events in memory [25] and may include information about specific sensory cues (e.g., visual or gustatory properties of a candy), affective values (e.g., pleasant sensations when we eat a candy), motivational properties (e.g., the satiation and nutritive impact) and specific response-eliciting characteristics (e.g., salivation) [26].

With the return of cognition in learning, modern learning theory overcomes the limitations of early learning models of AN, which were overly simplistic and have been justifiably criticised. Thus, the distinction between Pavlovian and instrumental conditioning is not only based on the type of events experienced and the experimental procedure used, but also includes what subjects learn (i.e., mental representations and learned associations in memory). On the other hand, unlike behaviourist positions, modern learning theory does not claim that anything can be learned or that all behaviour is learned, but rather the realisation that our biological systems and associative vulnerabilities constrain what we do or do not learn, promoting the learning of specific associations [27]. Surprisingly, progress in learning theory has not had a significant impact on clinical research and practice in EDs until very recently.

### How Cognitive Determinants Are Treated in Modern Learning Theory

Cognitive factors influence learning and performance in complex ways. In the first case, to the extent that learning is cognitively reconceptualised in terms of mental representations that are created, assembled and/or altered to better reflect the external environment [28], AN may be first characterised by the creation of unhealthy representations. Here, an example is the overvaluation of eating, weight and/or shape, which are considered to be the core psychopathology underlying AN [2]. In the case of the assembly of abnormal mental representations, an example may be the food-related phenomenon of thought–shape fusion, specific and distinct cognitive distortions present in patients with eating disorders. It occurs when thinking about eating high-caloric food leads individuals to feel fatter (e.g., “just thinking about eating a chocolate bar can make me gain weight”) [29]. An explanation advanced by modern learning theory is that activating the mental representation of sweet–fat foods will excite the feared consequences of eating as well, including the internal body sensations, via a link with the catastrophic weight gain representation. Likewise, given that one of the simplest forms of thought is an association in terms of mental representations of two events [30], maladaptive negative thoughts in AN (e.g., ”if I’m fat, people won’t like me”) may be understood as an association between two representations (fatness with social aversive experiences), resulting in exaggerated or pathological responses. Finally, with the introduction of cognition into learning, environment stimuli do not impose the content of learning mechanically on us; rather it opens new opportunities for an active role in the associative process. Thus, for instance, it has been suggested that people can acquire associations by engaging in rule-based processing based on language and formal reasoning [31].

In the second case related to performance, cognitive factors also influence responding; for instance, in the control of food-related behaviours [32]. Indeed, eating behaviour is often subject to sophisticated cognitive eating controls. One of the most widely practised forms of cognitive control over food intake is dieting, i.e., attempting to restrict intake as a means of weight regulation [33]. In AN patients, these cognitive regulations are especially important to overcome hunger sensations after long periods of deprivation. The problem is that anything that disrupts the cognitive control in people with a restricted diet (e.g., BP-AN) appears to unleash overeating [34]. Regarding the interplay between the cognitive content of learning and voluntary cognitive control processes in the context of food responses, both can be understood by a sequential pathway through a default-interventionist approach. Simpler automatic associative responses start and then high-level processes are recruited when the simpler responses prove inadequate, particularly when conflict is detected [32]. An example of conflict is when BP-AN patients refrain from their automatic tendency to eat attractive and pleasant chocolate in order to maintain incompatible goals in terms of weight status.

The main part of this review is focused on cognitive-associative learning rather than cognition-mediated performance mechanisms, with special emphasis on the creation, alteration and assembly of mental representations.

## 4. Progression of the Learning Models for Anorexia Nervosa

In early learning models, AN was seen as particular manifestations of an anxiety disorder. The assumption was that the pathological restriction of eating reduces anxiety (see [35,36]). This is well illustrated by the conceptualisation of AN as a weight phobia [37,38] in which patients limit their diet because they are anxious about weight gain. These theoretical models were mainly inspired by the two-factor fear theory (cf. [39]) combining Pavlovian and instrumental conditioning. Patients first showed a conditioned fear response through Pavlovian conditioning (Factor 1): caloric food-related cues occurring with weight gain (unconditioned stimulus) act as a warning stimulus (conditioned stimulus) of becoming fat, which elicits the anxiety/fear response (conditioned response). In a second phase, patients begin to diet and restrict their caloric food intake (action) in order to avoid weight gain and the conditioned fear response. Such avoidance via dieting is then negatively reinforced through anxiety reduction (Factor 2).

Ironically, although the two-factor model has had a major impact, it has never been directly tested for AN. Indeed, most of the studies in the conditioning basis of AN are descriptive and/or case reports (>65%; see Appendix A). There is only scarce and indirect evidence (see [35]). Moreover, within the two-factor model, it was difficult to explain why some patients with AN continuously restricted their calorie intake, even to the point of life-threatening starvation, and why certain patients, who wanted to restrict their calorie intake to lose weight, binge ate on a regular basis. Finally, the two-factor model itself underwent severe criticism (discussed elsewhere, [40,41,42]). As a result, the behaviourist learning perspective fell out of favour as a relevant model for AN.

There has recently been renewed interest in the anxiety-based model of conditioned avoidance for adults and adolescents with AN [43,44]. These new models still posit that avoidance behaviours are acquired responses with the aim of reducing eating-related anxiety. An innovation is that AN is now assumed to develop from vulnerabilities in emotional learning and memory processing. For instance, it has been proposed that patients with AN learn fear more easily than their healthy counterparts [45]. In addition, a wider range of conditioning experiences is now taken into account to explain how AN develops and is maintained, such as direct classical conditioning (e.g., food cues and traumatic experiences), verbal conditioning through information (e.g., threatening information about high-calorie food and overweight), vicarious conditioning (e.g., observing others with high-calorie food fears) and/or operant conditioning (e.g., when eating is followed by aversive consequences such as negative judgement from others or criticism) [46]. Likewise, other models based on learning processes such as the transdiagnostic theory for the treatment of eating disorders or the reward-centred model for the development and maintenance of AN have been described more recently (as discussed elsewhere; [47]). By contrast, the modern associative account of learning provides a much richer picture. For instance, abnormal behaviour is supposed to be activated not only via direct, instructional, verbal or vicarious pathways, but also by novel events that only share physical, perceptual or conceptual features with those representations currently maintained in memory, as well as by indirect, associatively retrieved representations of food stimuli (as observed using mediated learning paradigms; cf. [48], for a detailed discussion).

## 5. Introduction to a New Approach: Associative Analysis of Pathological Food Rejection

It is well established that patients with AN typically avoid the consumption of high-calorie foods vis-à-vis healthy individuals [49,50,51], even after completing treatment and restoring weight [52,53]. In order to understand why, modern associative learning theory provides the tools to conduct an experimental associative analysis of pathological food rejection, which requires identifying (1) the mental representations involved and (2) the associations established between these representations.

To do so, a number of methodological considerations should be taken into account. First, conditioning manipulations are critical tools to unravel the internal representational and associative processes [54,55] that underlie the pathological behaviour to be examined. Second, such changes in response after manipulations should be contrasted by multiple methods, including behaviours, self-reports and/or neural measures (e.g., according to the RDoC units of analysis; https://www.nimh.nih.gov/research/research-funded-by-nimh/rdoc/units/units-of-analysis; accessed on 21 January 2023). Third, experimental evidence examining acquired food rejection in a laboratory highlights two predominant associative processes: food avoidance motivated by conditioned fear and food aversion motivated by conditioned flavour aversions (see [56,57,58,59]). Therefore, it would be important to identify whether pathological food rejection in each AN patient is controlled by fear (i.e., related to external danger/threat), by flavour aversions (i.e., related to internal visceral discomfort) or both. Finally, to the extent that food rejection is conceptualised as an acquired avoidance behaviour, three types of associations may be expected to be involved: Pavlovian (between conditioned and unconditioned stimuli), goal-directed (between actions and outcomes, when an individual intentionally engages in actions that lead to a desired outcome) and/or habit (between stimuli and actions, when an action is automatically triggered by environmental stimuli) [40,41,42,60]. With these considerations in mind, let us now look at a detailed associative analysis of pathological food rejection in AN and the current empirical evidence.

### 5.1. Basic Processes Underlying Food Rejection Motivated by Fear

#### 5.1.1. Pavlovian Fear Reactions in Anorexia Nervosa

*What does associative learning theory have to say about food avoidance acquisition in AN?* Fear experiences are thought to play an important role in the onset and maintenance of maladaptive eating avoidance [61,62]. However, how do fear and food avoidance appear? From the species-specific defence reaction literature [63], an explanation posits that avoidance is rapidly acquired if the stimulus elicits defensive responses (e.g., fight or flight). For instance, spiders, heights or lightning elicit a range of innate defensive responses that phylogenetically predispose escaping from the potentially harmful situation. In contrast, “fleeing from” edible food neither belongs to the defence-reaction repertoire nor does it serve survival. In fact, the opposite is true. Food is essential for nutritional homeostasis and is a natural reinforcer that engages reward networks in the brain, with innate and learned appetitive reactions to caloric food. We argue that fear responses to edible food are counter-prepared to learn, and that intense unpleasant experiences (e.g., choking) are initially needed to reverse our natural appetitive reactions to caloric foods to unpleasant reactions, making the patients avoid them. (Nevertheless, the fact that food avoidance may also result from a weaker US in individuals with reduced appetitive reactions for food or in those that perceive strong rewards from food avoidance cannot be ruled out).

Another relevant question is how Pavlovian processes promote specific actions in AN such as abnormal food preferences and dysfunctional dietary patters. To do so, the Pavlovian-to-instrumental transfer (PIT) task offers a valuable, well-controlled procedure in which an ongoing instrumental action is enhanced by the presentation of a Pavlovian stimulus. In the PIT paradigm using food items, subjects typically undergo instrumental training in which one action earns a food outcome (A_1_ → O_1_) and another action earns a second food outcome (A_2_ → O_2_). In a separate Pavlovian phase, subjects learn that two stimuli differently predict those same food outcomes (S_1_ → O_1_ and S_2_ → O_2_). In the transfer test, the stimuli (S_1_ or S_2_) are presented while the subject freely chooses between the two actions (A_1_ or A_2_). What is observed is the PIT effect: each stimulus selectively primes the action that earns the same outcome, which is usually explained through the formation of specific S_1_–O_1_–A_1_ and S_2_–O_2_–A_2_ associative chains [64].

*Current findings from associative learning theory research*. In AN, an appetitive version of the PIT paradigm [65] investigated the impact of low-calorie and high-calorie food pictures on instrumental responses to these foods. During the Pavlovian phase, participants (mostly R-AN and healthy controls) received training during which one out of four cues (S_V_) predicted the display of a picture showing vegetables, while another cue (S_C_) predicted the display of a picture showing chocolate. Then, in the instrumental phase, the same participants were trained to press the letter “V” for vegetables (in Vogel et al.’s [65] paper, the letter related to vegetable was “G”: the first letter of the German word for “vegetable”. However, we replaced “G” with “V” here in order to make the reading easier) or “C” for chocolate to win vegetable/chocolate coins, thus receiving feedback on their performance. Finally, during the transfer test, participants were told that they could still earn vegetable-related or chocolate-related coins by pressing either “V” or “C” while the S_V_ or the S_C_ were also displayed in random order. A PIT effect occurred in aware participants who pressed the vegetable-related key “V” more often when the S_V_ had been presented compared with the presentation of the S_C_ or the neutral stimulus. As the S_V_ and pressing the letter “V” were never trained together, this PIT effect in the control of food response is explained by a chain of binary associations: S_V_–Vegetable–Pressing “V”. Unfortunately, no PIT research exploring the impact of pre-existing feared-conditioned stimuli on abnormal food choice, nor aversive PIT procedures with food as a threat and food avoidance response (cf. [66]) has been conducted in AN so far.

*What does associative learning theory have to say about food avoidance extinction in AN?* According to the associative structure of the PIT, if food stimuli selectively prime actions through the stimulus–outcome–action associative chain, it should be clear that the Pavlovian extinction of the stimulus–outcome association must reduce the expression of these actions (see Figure 3). Pavlovian extinction occurs when a conditioned stimulus is repeatedly presented alone. In clinical settings, extinction is known as exposure therapy, which means exposing patients to their feared food stimulus (e.g., sight and taste of foods) without the feared outcomes (e.g., weight gain) [44,46,67,68,69].

*Current findings from associative learning theory research*. Despite the efforts, limited evidence shows that in vivo food exposure decreases anxiety state and increases caloric intake and body mass index in AN (see [18,70]). For example, in one of the randomised controlled trials, Steinglass et al. [71] exposed AN patients to feared eating situations (e.g., holding a sandwich and eating it) without the use of anxiety-reducing rituals and safety behaviours (e.g., breaking it into small pieces) that prohibit successful extinction. Results showed a modest increase in intake (only 49 kcal from pre- to post-treatment during a test meal). The reason why Pavlovian extinction/exposure techniques usually result in only modest increases in caloric intake after exposure remains unclear and other mechanisms that are impervious to Pavlovian extinction have been proposed (see Section 5.1.3).

In summary, research on Pavlovian fear learning in AN is scarce. Although robust Pavlovian associations appear to be necessary in the acquisition of fear reactions that reverse the innate and learned preferences for high-caloric foods, more evidence remains to be gathered.

#### 5.1.2. Instrumental Goal-Directed Avoidance in Anorexia Nervosa

*What does associative learning theory have to say about the reinforcement control of food avoidance in AN?* Anorectic patients do not only refrain from eating, but also show active resistance to eating, including aggressive behaviour directed towards persons who try to interfere. Such eating patterns related to food restriction have long been considered as a form of instrumental action reinforced by consequences [72,73]. The question is whether food avoidance behaviour, and restrictive eating in particular, can be considered as an instrumental goal-directed action reinforced by consequences. If yes, it has to satisfy two criteria (see [74]). First, the individual must have the (implicit and/or explicit) knowledge of the causal relationship between the action and its consequences (belief criterion). Second, the expected consequences must be desired and, thus, have the status of a goal (desire criterion). These criteria seem to apply to patients with AN. For instance, (1) their primary belief is not to eat in order not to gain weight and (2) they desire not to gain weight (see [75,76]). One way to test the belief criterion includes changing the contingency between food intake and weight gain (e.g., showing patients that repeated food intake does not result in weight gain after weight restoration).

*Current findings from associative learning theory research*. The outcome-revaluation technique is another valuable conditioning tool to test the desire criterion. This consists of altering the value of the outcome: if an action (e.g., chocolate seeking) is controlled by the consequences (sweet chocolate), any change in the pleasant value of the goal (e.g., bitter chocolate) should affect that action. With this rationale, Godier et al. [77] used two outcome-revaluation paradigms in recovered R-AN and healthy women: a slips-of-action study with different fruit pictures (functioning as the stimuli and outcomes) and an avoidance task. No difference between the healthy participants and recovered R-AN patients was found, neither in the use of feedback to respond correctly to stimuli or in withholding responses for devalued outcomes in both paradigms. Notwithstanding, whether similar results would be obtained when it comes to R-AN-specific behaviours (e.g., responding to high-calorie food instead of fruits) remains to be explored.

*What does associative learning theory have to say about the original motivation for restrictive eating?* Lloyd et al. [62]; also, [69]) suggest that persons who develop AN begin to restrict their diet to reduce their fear of gaining weight. The initial motivation for restrictive eating might be the anxiolytic effect of dietary restriction (instrumental negative reinforcement). When restrictive eating leads to a loss of weight, it is positively reinforced by feelings of being in control, self-satisfaction and receiving compliments. A contrasting temporal course of the role of positive and negative reinforcement has been proposed by O’Hara, Campbell and Schmidt [78]: weight loss is perceived as a positive and rewarding outcome, promoting the development of anorectic behaviours (instrumental positive reinforcement). Thus, dieting may lead to fasting, which in turn results in the reluctance to gain weight and then to the aversive appraisal of food-related stimuli. Other authors have even suggested that the original reinforcers leading to weight loss might be represented by factors with no real interest for size, shape or body image distortion, such as the need for parental attention or preserving autonomy in children and adolescents [79,80]. Unfortunately, there is a lack of knowledge about the specific reinforcers playing a role and the temporal sequence in which both negative and positive reinforcements of food restriction may occur.

In summary, AN might involve complex, multi-step reinforcing phenomena to avoid food intake. Unfortunately, thus far, no study has assessed whether and when self-starvation symptoms in AN patients meet both the belief and desire criteria to be considered as a truly instrumental goal-directed action. In addition, the affective nature (unpleasant versus pleasant) of the initial and subsequent consequences that motivates dieting has not been systematically investigated.

#### 5.1.3. Habitual Avoidance in Anorexia Nervosa

*What does associative learning theory have to say about habitual responses in AN?* Habits typically refer to sequential, repetitive and motor actions elicited by stimuli that, once released, can go to completion without conscious oversight. Habitual actions are acquired over the course of time, becoming remarkably fixed. In AN, patients are often described as rigid, inflexible and perfectionistic [81], as well as engaging in fixed behavioural patterns regarding the purchasing, preparation and consumption of food. Even after receiving treatment aimed at normalising weight and eating patterns, patients with AN continue to consume fewer total calories and fewer calories from fat than their healthy peers [52]. These characteristics may reflect a tendency to develop repetitive, stereotyped behaviours, and a vulnerability to forming strong aberrant habits in these patients’ daily lives [82]. Habits represent the second type of instrumental action, involving stimulus → action associations. Note that their associative structure does not include the outcome or the unconditioned stimulus. Therefore, habits are insensitive to outcome/unconditioned stimulus revaluation and extinction techniques: they involve stimulus-elicited reactions without the retrieval of unconditioned stimulus [12].

*Current findings from associative learning theory research*. There are compelling behavioural and neural data to suggest that habitual processes may underlie the persistence of AN (see [83], for a review). Especially in the later stages when illness becomes more persistent, restrictive eating has been suggested to be a compulsive, habitual and entrenched behaviour [62,72,78,82,84]. For instance, habit strength as measured by the Self-Report Habit index has been found to be a significant predictor of self-reported food restriction [85], as well as associated with the duration of illness in AN patients [86]. Using fMRI and functional connectivity analysis, Foerde, Steinglass, Shohamy and Walsh [87] examined pre-existing maladaptive food choices with high-fat and low-fat options in chronic AN women and healthy women. The authors found higher engagement of the dorsal striatum in AN women than in healthy controls when making restrictive choices about what to eat. In addition, AN women showed greater connectivity in fronto-striatal circuits for low-fat than for high-fat foods (whereas healthy controls showed the opposite pattern). Given that fronto-striatal networks are also important for the development of habitual behaviour, the authors concluded that dietary behaviour in the repertoire of chronic AN was controlled by habitual processes.

From unhealthy dieting to chronic restriction and starvation, there has been increasing recognition of different illness stages along the life course in AN (cf. [88]). For example, Treasure et al. [88] suggest that early stage illness for AN should be defined as an illness duration of ≤3 years as clinical outcomes become poorer once the illness duration exceeds 3 years. This suggestion is consistent with the finding that, after years of dieting and weight loss, patients report the gradual loss of control that occurs as rigid restrictive eating and starvation become highly resistant to change. Whether the transition from early to chronic dietary restriction in AN reflects a shift from goal-directed (controlled by action → outcome associations) to habitual (controlled by stimulus → action associations) behaviours still remains an intriguing question. If true, interventions designed especially for children/younger adolescents and implemented as early as possible should receive more attention in order to prevent the transition to chronic, treatment-resistant eating habits.

### 5.2. Basic Processes Underlying Food Rejection Motivated by Taste Aversion

*What does associative learning theory have to say about food aversion?* Although several learning processes cause avoidance behaviours, taste aversion is a unique category of avoidance. For example, if after consuming a new dish for the first time one person suffers visceral discomfort with nausea and disgust while another person responds with an allergic reaction, then both individuals will avoid eating that dish in the future. Only the first person in this example, however, will develop a strong distaste and taste aversion. The second person will not consume this dish again to prevent another fearful allergic reaction, but the taste will remain unchanged.

Acquired taste aversions arise specifically from exposure to stimuli that produce nausea and a qualitative shift in palatability. They are the product of a hard-wired system connecting the nose and mouth, gastrointestinal tract and brain that allows animals to learn about toxic foods and limit intake. Interestingly, food or taste aversions have the potential to overrule the biological urge to eat and drive the restriction of food. Subsequent to this affective (subconscious) process, taste is further integrated with other sensory attributes of the flavour, such as odour or texture. Together, these affective and cognitive processes yield an adaptive system that enables organisms to learn which foods are safe to eat and which are not [89].

As is clear from this example, the amount of food is not sufficient to reveal whether food avoidance is due to fear- or aversion-related processes. Thus, traditional measures such as consumption suppression are insufficient and additional measures such as facial expressions and neurobiological dissociations are needed. For example, fear and flavour aversions elicit responses that are differentiated by unique facial expressions [90]. A fearful expression includes widening eyes, raising eyebrows and flaring nostrils, while aversion-related disgust reactions are characterised by a lowered brow, closed eyes and scrunched-up nose. Of interest is the clinical observation that one of the most frequent responses of AN patients to the question “what is the worst consequence of eating?” is the fear of feeling disgusted [46]. People with AN often experience disgust with respect to sexuality, parts of their bodies and towards certain foods, especially those which are fattening or have a high calorie content (cf. [91]).

*Current findings from associative learning theory research*. When facial expression analysis is applied to patients, reduced pleasantness as measured by the decreased activity of zygomatic muscles to food cues has been observed in R-AN compared with healthy controls [92]. In addition, levator labii muscle activation (a reliable index that appears to be unique to the emotion of disgust) has been reported during a food-based reversal learning task using neutral stimuli and pictures of chocolate candies for R-AN, compared with young control females [93]. Regarding neuroimaging studies, the predictable administration of sweet stimuli has been shown to be associated with reduced activation in the taste–reward regions of the brain in individuals with AN (e.g., insula, ventral and dorsal striatum) (see [94] for a review). Interestingly, AN patients do not appear to show an increased global disgust sensitivity but only one that is specific to areas that concern food and the body [95].

#### Disgust and Flavour Aversions in Eating Disorders

*What does associative learning theory have to say about food aversion acquisition in AN?* It has been argued that food may acquire disgust-eliciting properties and become intrinsically revolting in AN [96,97,98]. Such conditioned flavour aversions may be promoted by gastrointestinal disturbances in AN. For example, gastrointestinal disturbances are common and develop along with the disordered eating behaviour and the ensuing malnutrition and subside with the resumption of normal food intake and body weight [99,100]. Another source may be the experiencing of the visceral effects of restricted food access along with intense exercise, as seen in healthy animals where AN-like behaviours (e.g., self-starvation and hyperactivity) result in acquired aversions to a preferred food [101].

Considering psychological sources, retrospective reports in AN patients suggest that flavour aversions may be elicited by the knowledge of or imagining disgusting pictures without any physical illness at all [102]. In addition, expectations about the impact of food on the body (“becoming fat”) resulting in body-related self-disgust have been suggested [91]. These cognitive aversions appear to be more frequent and stronger than in healthy individuals [103], more likely to generalise to other foods and more resistant to extinction than physical aversions [102]. Furthermore, it has been argued that dysfunctional thoughts about body/weight (e.g., “this food increases body weight” or “the mere thinking of food may increase weight”; [104]) may be able to make one feel bad while eating and these negative feelings to extend to the affective value of taste, making high-calorie food taste worse and resulting in the early termination of intake [105].

*Current findings from associative learning theory research*. Lascelles, Field and Davey [106] demonstrated that the negative evaluation of a body image can be transferred, through a process of associative learning, to food with which the body image has been paired in healthy women. This resulted in a negative affective shift for those foods. This possibility might also help to explain why many individuals who internalise thin ideals of the body or who have had experiences of humiliation or sexual abuse are at risk of developing an ED (cf. [107]). A hypothesis is that they might develop food aversions mediated by cognitive disgusting images, especially by disgusting images about their own body. In any case, whether flavour aversions are a result rather than a cause of developing AN remains to be ascertained, as does the specific gustatory, physiological and/or cognitive event responsible for such acquired flavour aversions.

### 5.3. Multiple and Different Associations to Explain Anorexia Nervosa Subtypes

We posit that the number and type of associations are critical to distinguish AN subtypes and to explain changes in symptomatology in a patient over time. For instance, given that “intense fear of gaining weight or becoming fat, even though underweight” is one of the diagnostic criteria of AN [2], both BP-AN and R-AN may be accordingly characterised by the presence of a Pavlovian feared sweet–fat food cue → catastrophic weight gain association that promotes dieting (see Table 2). Furthermore, in order to overrule the physiological needs of a state of energy deficit and, therefore, explain severe undernourishment in R-AN, a second aversive association that concurs in the same restraint direction is assumed: sweet–fat flavour → gastrointestinal malaise. By contrast, binge eating, compulsive eating and even overeating might be explained in BP-AN patients by a different type of second association: sweet–fat flavour → enhanced pleasant gustatory sensations. Indeed, bulimic patients report a higher liking for high-sweet stimuli [108] and heightened preferences for sweets [109] vis-à-vis healthy controls, but without differences for low-sweet solutions [110]. In such cases, unlike R-AN, bulimic spectrum disorders (including BP-AN patients) would have encoded in their memory two conflicting associations (fear versus appetitive): sweet–fat flavour predicts a delicious taste and catastrophic weight gain. This would be responsible for recurrent episodes of restriction and binging and the important level of ambiguity observed at an emotional level (cf. [111]).

In order to resolve the ambiguity and the behavioural approach–avoidance conflict, information provided by other cues (e.g., the context) is expected to be used (see [112], for a review). For instance, AN patients develop fear of becoming fat in situations predicting caloric eating such as the kitchen or mealtime. Then, these contextual cues should further excite the fear association of sweet–fat cues with weight gain (Figure 4), promoting a high-arousal state that inhibits motivation to eat and food intake, as well as cognitive eating controls, to further maintain food rejection beyond physiological needs [113]. However, in BP-AN, sweet–fat flavours are embedded not only in associations that excite the memory of the feared postingestive consequences of eating, but also in associations that serve to activate the memory of pleasant sensations of sweetness and hunger-reduction. It should be noted that preference for sweet food (which could also still be present premorbidly since food is rewarding for most people) is usually enhanced under food deprivation conditions. Then, the presence of interoceptive hunger cues will heighten the ability of sweet–fat flavours to retrieve the attractiveness of sweetness and overconsumption. In the absence of hunger cues, by contrast, this appetitive association should be reduced, making it more likely that these food-related cues will retrieve the memory of gaining weight, thereby reinstating food avoidance and food restriction. Thus, food restriction or binge eating will result in competition between the relative activation of feared and appetitive associations in each context.

By considering the type and number of associations, this approach can explain why some patients restrict their calorie intake to the point of life-threatening starvation and others restrict it to lose weight and binge on a regular basis. The development of additional associations might also explain symptoms fluctuations and migration, transitions across ED diagnostic categories and the fact that, for example, two patients with the same diagnosis can display restrictive or binge-eating episodes across different contexts.

Moreover, the restrictive and binge–purge AN subtypes may be separated by specific associative features, such as differences in instrumental goal-directed actions. Based on results from studies using de novo conditioning in the laboratory, it seems that the goal-directed system is relatively intact in R-AN patients (although it might be affected in the BP-AN group). Indeed, deficits in goal-directed tasks with general (money) and illness-specific (food) outcomes have been found for patients with BP-AN but not for patients with R-AN [114,115]. This is consistent with reductions in goal-directed learning found in other binge-eating groups such as bulimia nervosa and binge-eating disorders [116]. By contrast, the development of trained goal-directed behaviours does not appear to differ between patients suffering from R-AN and healthy participants, as reported by the outcome-revaluation studies as well as the lack of differences in the acquisition and expression of food-related instrumental goal-directed responding between these groups during the PIT tasks [65].

## 6. Abnormal Associative Processes and Vulnerability in Anorexia Nervosa

Associative processes only acquire a pathological role when interacting with premorbid vulnerabilities and precipitating factors. In associative terms, vulnerabilities might be explained by the abnormal formation of links between mental representations. Put simply, patients might show a lowered/stronger tendency to form/weak associations than their healthy counterparts (see [117]). In this way, factors ranging from genetic to sociocultural ones may be translated into associative vulnerabilities, i.e., the abnormal acquisition and/or extinction of learned behaviours.

The lack of studies on associative vulnerabilities in AN represents a noticeable gap in the literature, despite several lines of evidence supporting this notion. In fear learning, these include hyperresponsivity to food in fear circuits (i.e., amygdala) and body cues in underweight phenotypes [118], as well as rapid fear-based learning and slower rates of fear extinction to calorie-dense foods [19,119]. Interestingly, young women with high scores on ED symptoms have been shown to be characterised by a heightened proneness to associate disgusting outcomes with food [120]. In this study, female undergraduate students with high and low ED symptomatology participated in a Pavlovian disgust-conditioning procedure in which one of two neutral foods (S+) was followed by videos depicting people vomiting while the other one was not (S–). Finally, both Ss were presented on their own during extinction. Only the high ED group considered the S+ as more disgusting and fear inducing and were less willing to eat compared with the S- after both acquisition and extinction. Interestingly, higher disgust was associated with reduced calorie consumption over a 24 h period across groups.

In the single experimental study using disgust conditioning with patients to date, Hildebrandt et al. [93] used a food-based reversal learning task with acutely ill R-AN adolescents in which a picture of chocolate candies was associated with stimulus A (S_A_) in the first phase, but not with stimulus B (S_B_). In the second phase, the contingencies changed without warning and S_B_ was paired with the picture of chocolate candies, while S_A_ was always presented alone. S_A_ and S_B_ were two different coloured squares. The results showed that disgust responses to S_A_ predicted more difficulty in reducing the association between S_A_ and candies (impaired Pavlovian extinction) in R-AN, but not in healthy controls. Moreover, impaired extinction was evident at the onset of the illness, i.e., before chronic starvation and brain development had had the opportunity to contribute to this deficit. This strongly suggests impaired extinction learning as a risk factor for AN.

## 7. Future Directions and Clinical Implications

We will now focus on several questions that remain unanswered and how they should be addressed, aspiring to guide future research.

*Are conditioned flavour aversions causally related to food restriction in AN?* Although there is provoking evidence that disgust reactions and flavour aversions to forbidden foods (and related stimuli) may mediate extreme food avoidance in AN patients, more research is needed in order to ensure that disgust is not merely a co-existing phenomenon of the food restriction. In order to test such a possibility, pharmacological manipulations may be considered to reduce anticipatory nausea and food aversions. For instance, the extent to which the rejection severity of sweet products is alleviated by the administration of antiemetic drugs, as observed in other anorectic patients (e.g., chemotherapy-induced anorexia) who are nauseous or with learned food aversions [121], might be examined.

*How to reveal the content of learning in AN?* It is crucial to conduct assessments of the pathological content of learning in AN. By content of learning we mean the types of mental representations and associations. To do so, we have highlighted two strategies. The first type of strategy is to examine the consequences of altering the relationship between events, such as interrupting the contingency between the conditioned and unconditioned stimuli in Pavlovian extinction or between the action and the outcome in instrumental extinction. The second type uses tests of various sorts, such as the Pavlovian-to-Instrumental transfer test and outcome devaluation (see Figure 3). In this sense, additional strategies may also be suggested. For example, focused on the mental representations that are acquired in AN, Murray, Loeb and Le Grange [122] have provided food intake or weight normalisation as the conditioned stimuli, and weight gain, embarrassment or social rejection as the unconditioned stimuli. Surprisingly, these suggestions are yet to be examined. Interestingly, superconditioning, blocking or transreinforcing blocking effects might be useful tools as a first step in the diagnosis of mental representations. Blocking [123] refers to the observation that a first conditioned stimulus (S_1_) that already predicts an unconditioned stimulus often reduces conditioning to a second neutral stimulus (S_2_) when both stimuli are paired again with the same unconditioned stimulus. When transferred, if the food cue is already acting as a S_1_ that predicts weight gain in AN (as suggested by Murray et al., [122], then food should block fear learning about a novel food cue (S_2_) when both are paired with weight gain again (see [124], for a recent demonstration and review of the blocking effect).

Likewise, a broader range of conditioning procedures (e.g., changes in the outcome magnitude or in the motivational state relevant to the outcome) in combination with neural systems analysis (see [125], for an example of how neural analysis may help us to understand the contents of learning) and psychobiological techniques (e.g., transcranial magnetic stimulation following exposure to food cues; Rachid [126]) are available to help determine the mental representations and associations that promote food restriction in AN. For instance, transcranial magnetic stimulation could be used to temporarily inactivate a particular brain area associated with the neural pathways responsible for fear (e.g., ventromedial prefrontal cortex) or disgust (e.g., anterior insula) in order to distinguish the specific role of fear versus flavour aversions. Additionally, we could triangulate data using specific measures. Thus, whether pathological food restriction is driven by flavour aversions could be confirmed by pairing forbidden foods to neutral flavour stimuli via classical conditioning procedures and testing the acquisition of new aversions to these initially neutral flavour stimuli by disgust-related orofacial muscular activity.

*How can patient-tailored treatments be enabled?* Each treatment should address the specific pattern of abnormal associations in each patient. For instance, if extreme food avoidance is motivated by flavour aversions, the novel target of reducing aversive reactions and anticipatory nausea should be included. This could be obtained through the development of new conditioned food preferences for “forbidden foods”. Likewise, exposure therapy should not target the extinction of fear (as currently recommended in traditional cognitive-behavioural or family-based treatments [127]), but the extinction of disgust. It should be noted that both learned emotional reactions, conditioned fear and disgust, are partially independent of each other [128,129] and, therefore, the extinction of one should not affect the other.

*How can we boost additional interventions in order to create new healthy associations?* Interventions based on learning and experience-dependent plasticity to rewire the brain’s associations beyond traditional approaches are needed (e.g., [130]). In particular, it is critical to promote faster, stronger and more durable associations to overcome the original pathological associations. An example is provided by the d-cycloserine-augmented psychological therapies, including the use of cognitive enhancers to boost the development of new healthy associations during exposure therapy in AN [67]. In those with a chronic and unremitting course of the disorder, habit-centred approaches (i.e., Regulating Emotions and Changing Habits; [131]) based on stimulus → action rather than action → outcome associations may also offer a new way forward. Since habitual avoidance is extremely resistant to extinction, current clinical alternatives focus on suppressing the habit by making the performance of the habit impossible, removing situations/stimuli that activate or trigger the habitual behaviour or exerting a top-down control of habits after exposure to the cue that has activated the reaction in the memory (Figure 3) [132].

*What is the role of each food attribute (orosensory, postingestive and ideational) in AN?* Food choice and food intake are guided differently by sensory and by metabolic processes. Moreover, it is well established that food learning that promotes food consumption and food avoidance may be mediated by associations with other tastes (flavour–taste learning) or calories (flavour–nutrient learning) [133,134]. Unfortunately, the specific contribution of each food component in AN symptomatology has been largely ignored by clinical psychology and psychiatry. For instance, regular cognitive-behaviour therapy for eating disorders usually overlooks the fact that people suffering from AN might have trouble recognising tastes or responding to the pleasure associated with food during food exposures. Likewise, their role in the ambivalence towards high-calorie food is unknown in BP-AN patients showing enhanced pleasant, sweet perception and binge eating, but calorie fear driving food avoidance (e.g., highly restrictive eating, strict eating rules, body checking) and purging behaviour (e.g., vomiting) if those caloric foods are actually eaten.

*Obtaining the whole picture beyond basic associative learning structures*. Finally, although the comprehensive discussion of associative content underlying food restriction in AN exceeds the scope of this review, other associative structures may be anticipated, in particular, if the context is taken into account. Indeed, we can propose the potential role of serial conditioning or second-order conditioning (in which patients learn the associative chain S_1_ → S_2_ → outcome), occasion setting (with the associative structure S_1_: S_2_ → outcome, where the S_2_ → outcome is valid only under the circumstance S_1_) or even hierarchical structures involving avoidance responses [135,136,137,138,139]. One example of the role of the context is the limited effectiveness of the incentives used in the inpatient setting during posthospitalization. Indeed, the high rates of relapse after hospitalization suggest that inpatient behavioural treatment may be context-dependent. Modifying specific AN behaviours often neglects the contextual circumstances, which is often inadequate to foster lasting changes in eating patterns to sustain normal weight, e.g., when the external structure of the hospital program is removed [44]. Let us look at another example with a restaurant as the context. From a serial conditioning point of view, a patient could be scared when going to a restaurant (S_1_) as it is associated with food (S_2_) and food, in turn, with weight gain (outcome). Conversely, from an occasion setting perspective, the restaurant (S_1_) might set the occasion for the patient to be scared of food: while many AN patients have no problem seeing or even cooking food at home, the restaurant would be a context where they cannot avoid food intake and, thus, weight gain (S_2_ → outcome). Importantly, different interventions will be necessary if the restaurant is a serial conditioned stimulus or an occasion setter. To our knowledge, however, no previous research under traditional treatment orientations has been conducted on exposure to contexts (such as a restaurant or family dinner) in AN patients.

## 8. Conclusions

The associative learning framework may provide a major step in advancing our understanding of food restrictive patterns at the conceptual and methodological levels. It is true that little work has been conducted to investigate and disentangle what patients have learned in associative terms, despite their important implications for aetiology, case conceptualisation and intervention. Therefore, the relative absence of human clinical data from associative learning experiments provides ample opportunities for future research in this area to translate basic behavioural findings into changes in practice. We expect more research on associative learning to tailor specific prevention and intervention strategies to the needs of individual AN patients. More than 50 years of animal and human research in modern learning theory, with a level of sophistication in psychological theorising and experimental methodology hardly seen in the preceding century of studies on learning and behaviour, should enable us to successfully improve AN clinical practice.

## Figures and Tables

**Figure 1 behavsci-13-00096-f001:**
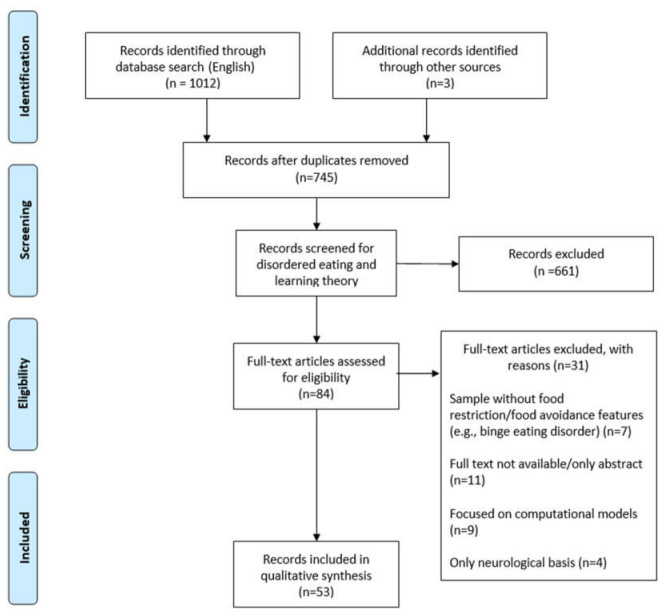
PRISMA 2009 flow diagram and selection of original articles. Moher, D.; Liberati, A.; Tetzlaff, J.; Altman, D.G. Preferred reporting items for systematic reviews and meta-analyses: the PRISMA statement. *PLoS Med*. **2009**, 6, e1000097. https://doi.org/10.1371/journal.pmed.1000097.

**Figure 2 behavsci-13-00096-f002:**
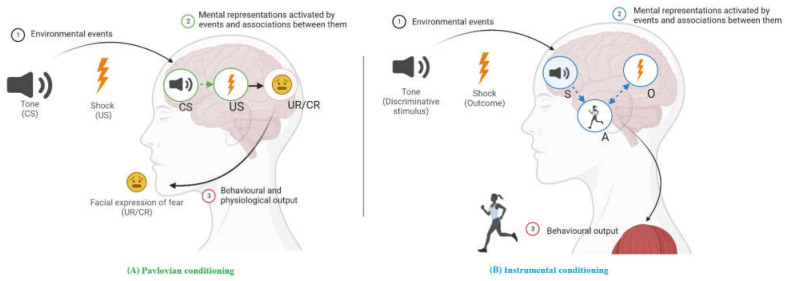
Content of Pavlovian (**A**) and instrumental (**B**) learning showing the mental representations and associations acquired after the conditioning experience. (**A**)|Pavlovian conditioning is viewed as involving conditioned stimuli (CS) and unconditioned stimuli (US), such as the pairing of a tone with an electric shock. These pairings result in a CS→US association in memory (in green) through which the tone elicits fear responses such as facial expressions of fear. (**B**)|Instrumental conditioning (in blue) in which a response (e.g., jumping) is followed by an outcome (e.g., an electric shock) and results in an action–outcome (A→O) association. After many repetitions, a new habitual stimulus–action (S→A) association is formed, such as between the tone and jumping. Note: Circles represent mental representations in memory. Lines suggest how one can influence another: solid lines indicate innate links and dashed lines indicate links that can be strengthened or weakened by experience. Activation is shown by an arrow. A: instrumental action; CR: conditioned response; CS: conditioned stimulus; O: outcome; S: discriminative stimulus; UR: unconditioned response; US: unconditioned stimulus. Created in Biorender.com.

**Figure 3 behavsci-13-00096-f003:**
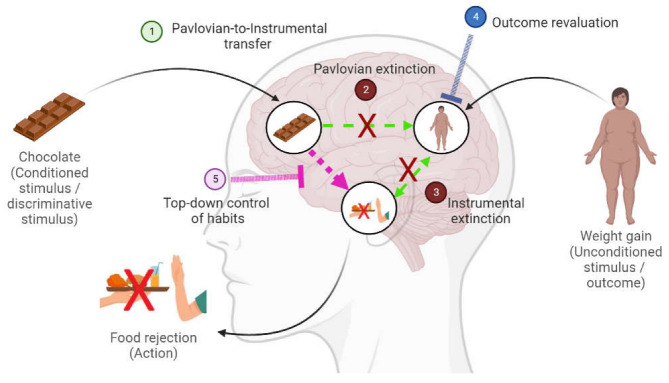
Examples of learning procedures to test and modify mental representations (e.g., outcome revaluation; in blue), associations between conditioned and unconditioned stimuli (e.g., Pavlovian extinction; in brown) or between actions and outcomes (e.g., instrumental extinction; in brown), associative chains (e.g., Pavlovian-to-Instrumental Transfer task; in green), and/or habitual stimulus–action associations (e.g., top-down control of habits; in pink) in pathological food rejection motivated by fear. (1) According to Pavlovian-to-instrumental transfer, a stimulus (chocolate) may promote instrumental avoidance (food rejection) through its link with the outcome (gaining weight) via the associative chain stimulus–outcome–action. (2) During Pavlovian extinction, the conditioned stimulus (chocolate)–unconditioned stimulus (weight gain) association is compromised and the ability of that stimulus to activate the unconditioned stimulus is reduced, decreasing the expression of fear responses. (3) In instrumental extinction, the association of an action (food rejection) with an outcome (weight gain) is degraded. (4) Outcome revaluation consists of altering the value of the outcome mental representation (weight gain). (5) A second pathway to evoke instrumental food rejection is through direct association between the discriminative stimulus (chocolate) and the action of rejecting food in the way of habits. This association is not affected by Pavlovian/instrumental extinction or outcome revaluation procedures given that habits are independent of the associations with the unconditioned stimulus/outcome (weight gain). Note: Conventions and abbreviations are as given for Figure 2. Created in Biorender.com.

**Figure 4 behavsci-13-00096-f004:**
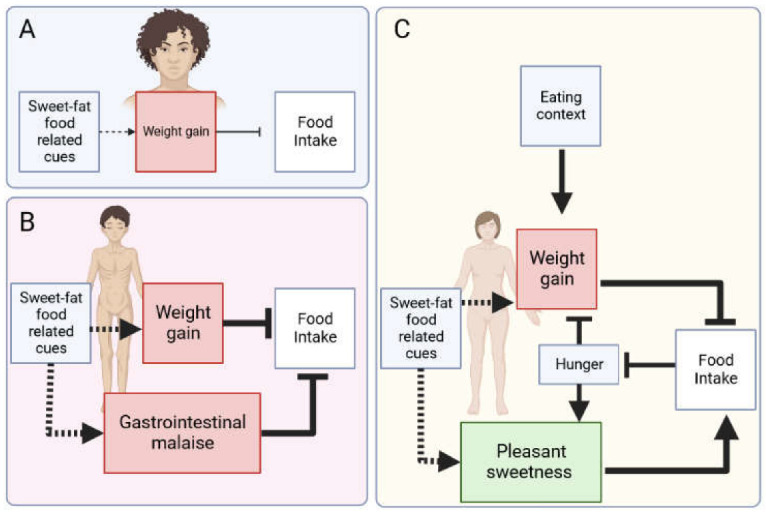
Some possible associations for people at risk of developing anorexia nervosa (**A**) or suffering from anorexia nervosa restricting type (**B**) or anorexia nervosa binge-eating type (**C**). (**A**)|Fear of gaining weight is common all across their lifespan for many women, promoting dieting, which is a risk factor for developing an eating disorder such as anorexia nervosa and bulimia nervosa. (**B**)|Anorexia nervosa restricting type exhibits two aversive associations, promoting extreme food restriction in which sweet–fat food-related cues (e.g., sweet taste) are linked to fear of gaining weight and to disgusting visceral malaise. (**C**)|Anorexia nervosa binge-eating type is characterised by two associations with opposed motivational value (aversive in red and appetitive in green), which promote food restriction (related to fear of weight gain) or food intake (related to pleasure). As a result, the relationship with food-related cues is ambiguous, and patients must depend on the presence of other cues to resolve this ambiguity, such as hunger or situations that predict eating. Note: Lines suggest how one can influence another; dashed lines indicate associations with sweet–fat food-related cues that can be strengthened or weakened by experience. Activation is indicated by arrows and inhibition by bar-headed lines. The width of the arrows and the font size variation symbolise the intensity of associations and mental representations, respectively. Created in Biorender.com.

**Table 1 behavsci-13-00096-t001:** Strategy and term combination for the search in databases.

Search	Strategy	Descriptors and Keywords
#1	Focused on behaviourist learning theory and eating disorders	Descriptors for learning (“learning theory” OR “reinforcement” OR “stimulus-response” OR “classical conditioning” OR “operant conditioning” OR “instrumental conditioning” OR “respondent conditioning” OR “Pavlovian conditioning” OR “instrumental learning”) AND descriptors for eating disorder (“anorexia nervosa”)
#2	Focused on modern learning theory and eating disorders	Descriptors for learning (“associative learning” OR “content of learning” OR “modern learning”) AND descriptors for eating disorder (“anorexia nervosa”)
#3	Focused on eating behaviour and conditioning	Descriptors for learning (“conditioning theory” OR “conditioning procedure” OR “conditioning learning”) AND descriptors for eating behaviour (“eating disorder” OR “disordered eating”)

**Table 2 behavsci-13-00096-t002:** Possible associative differences to explain subclinical populations of anorexia nervosa (AN), including the restricting type (R-AN) and binge–purging type (BP-AN), and healthy people.

Dysfunctional Association in Memory	R-AN	BP-AN	No Patients	Process	Outcome
Sweet–fat food-related cue → Catastrophic weight gain	✓	✓	×	Conditioned fear learning	Dieting, food restriction
Sweet–fat flavour → Gastrointestinal malaise	✓	×	×	Conditioned flavour aversion learning	Food avoidance
Sweet–fat flavour → Pleasant gustatory sensations	×	✓	×	Enhanced learned flavour preference	Overconsumption, binge eating

## Data Availability

Data are contained within the article or Appendix A.

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
