# Peer review of "Food Restriction in Anorexia Nervosa in the Light of Modern Learning Theory: A Narrative Review"

_behavsci, 2023, doi:10.3390/bs13020096_

Round 1

Reviewer 1 Report

I commend the authors for the well-structured, comprehensive, and useful review that will significantly aid a large contingent of psychiatrists trying to understand the neurological processes of food restriction phenotype in anorexia nervosa. The discussion of the future directions and clinical implications is highly appreciated.

I have a few minor issues:

1. The most intriguing part is applying your theoretical framework to distinguish AN subtypes, for example, the part 5.3, “Multiple associations and anorexia nervosa subtypes”. I believe adding a summarized figure/illustration can strengthen the manuscript and make it more readable.

2. The abstract is too generic and hard to capture the key advantages of your theory framework, I suggest to add/modify a few sentences to clarify how the associative learning theory improves our mechanistic understanding and clinical managements.

3. May be worth one more proofreading session to double check some few typos or grammar issues. For example:

1). Line 13, “the search for innovative approaches continuous at the laboratory and clinical levels”, “continuous” should be “continues”.

2). Line 23, “Nonetheless, it is strike…”, “strike” should be “striking”.

3). Line 24, “how little impact modern learning theory have had on current AN research agenda and practice”, “have had” should be “has had”.

4). Line 125, “Proponents of behaviourist learning theory intentionally ignored”, “behaviourist” should be “behavioural”.

Author Response

Review #1

I commend the authors for the well-structured, comprehensive, and useful review that will significantly aid a large contingent of psychiatrists trying to understand the neurological processes of food restriction phenotype in anorexia nervosa. The discussion of the future directions and clinical implications is highly appreciated.

I have a few minor issues:

  1. The most intriguing part is applying your theoretical framework to distinguish AN subtypes, for example, the part 5.3, “Multiple associations and anorexia nervosa subtypes”. I believe adding a summarized figure/illustration can strengthen the manuscript and make it more readable.

This section has been recast and a new figure (Figure 4) has been added to make it more readable (p.14):

“In order to resolve the ambiguity and the behavioural approach-avoidance conflict, information provided by other cues (e.g., the context) is expected to be used (see Bouton, 1993, for a review). For instance, AN patients develop fear of becoming fat in situations predicting caloric eating such as the kitchen or mealtime. Then these contextual cues should further excite the fear association of sweet-fat cues with weight gain (Figure 4), promoting a high-arousal state that inhibits motivation to eat and food intake, as well as cognitive eating controls, to further maintain food rejection beyond physiological needs (Macht, 2008). However, in BP-AN, sweet-fat flavours are embedded not only in associations that excite the memory of the feared postingestive consequences of eating, but also in associations that serve to activate the memory of pleasant sensations of sweetness and hunger-reduction. It should be noted that preference for sweet food (which could also still be present pre-morbidly since food is rewarding for most people) is usually enhanced under food deprivation conditions. Then, the presence of interoceptive hunger cues will heighten the ability of sweet-fat flavours to retrieve the attractiveness of sweetness and overconsumption. In the absence of hunger cues, by contrast, this appetitive association should be reduced, making it more likely that these food-related cues will retrieve the memory of gaining weight, thereby reinstating food avoidance and food restriction. Thus, food restriction or binge eating will result in competition between the relative activation of feared and appetitive associations in each context”.

  1. The abstract is too generic and hard to capture the key advantages of your theory framework, I suggest to add/modify a few sentences to clarify how the associative learning theory improves our mechanistic understanding and clinical managements.

According to the reviewer’s suggestion, the abstract has been modified accordingly:

“Improvements in clinical management of anorexia nervosa (AN) are urgently needed. To do so, the search for innovative approaches continues at laboratory and clinical levels to translate new findings into more effective treatments. In this sense, modern learning theory provides a unifying framework that connects concepts, methodologies and data from preclinical and clinical research to inspire novel interventions in the field of psychopathology in general, and of disordered eating in particular. Indeed, learning is thought to be a crucial factor in the development/regulation of normal and pathological eating behaviour. Thus, the present review not only tries to provide a comprehensive overview of modern learning research in the field of AN, but also follows a transdiagnostic perspective to offer testable explanations for the origin and maintenance of pathological food rejection. This narrative review was informed by a systematic search of research papers in the electronic databases PsycInfo, Scopus and Web of Science following PRISMA methodology. By considering the number and type of associations (Pavlovian, goal-directed or habitual) and the affective nature of conditioning processes (appetitive versus aversive), this approach can explain many features of AN, including why some patients restrict food intake to the point of life-threatening starvation and others restrict calorie intake to lose weight and binge on a regular basis. Nonetheless, it is striking how little impact modern learning theory has had on the current AN research agenda and practice”.

  1. May be worth one more proofreading session to double check some few typos or grammar issues. For example:

1). Line 13, “the search for innovative approaches continuous at the laboratory and clinical levels”, “continuous” should be “continues”.

2). Line 23, “Nonetheless, it is strike…”, “strike” should be “striking”.

3). Line 24, “how little impact modern learning theory have had on current AN research agenda and practice”, “have had” should be “has had”.

All these typos and grammar issues have been revised and an additional proofreading session to double-check the manuscript has been carried out.

4). Line 125, “Proponents of behaviourist learning theory intentionally ignored”, “behaviourist” should be “behavioural”.

The term “behaviourist learning theory” has been used instead of “behavioural learning theory” in order to stress the approach to psychology, formulated in 1913 by John B. Watson (see American Psychological Association Dictionary of Psychology, 2022), based on the study of objective, observable facts rather than subjective, qualitative processes, such as feelings, motives, and consciousness. Behavioural learning models may also include other approaches such as the social learning theory developed by Albert Bandura. Nevertheless, we will replace “behaviourist” by “behavioural” if the reviewer insists. 

Reviewer 2 Report

The present study reviewed and provided a potential mechanism for AN based on modern learning theory. The authors argued the development of AN could begin with classical conditioning, but the instrumental goal-directed avoidance conditioning reinforced the patients with AN to restrict food intake. Last, denying food could be a habit for patients with AN.

Despite these inferences sounding like they make sense, some comments could be considered further:

First, the determinant of cognitive factors seems to need to be considered even though memory and re-appraisal are information processing components. Therefore, the authors needed to explain how cognitive determinants will be treated in modern learning theory. 

Second, how does modern learning theory explain subclinical populations of AN? Please have more illustrations.

Third, it will be better to provide a framework of modern learning theory for AN in the article. The authors could have a table or figure to explain the mechanism of AN.

Forth, please rewrite the abstract. In this abstract, I needed help understanding the purposes of the present study. Do learning theory and food restrict, which is the point of this review? Besides, what were possible explanations of the role of food restriction in AN based on learning theory? And how many articles were reviewed in the present study? Please revise this abstract and make it more concrete.

Thanks for considering my comments; I hope these comments are helpful to you.  

Author Response

Dr. Lana Tucakovic

We are resubmitting a new version of our manuscript. The manuscript has been revised according to the suggestions and comments of the reviewers. In what follows I list the reviewer's comments and answers.

Review #2

The present study reviewed and provided a potential mechanism for AN based on modern learning theory. The authors argued the development of AN could begin with classical conditioning, but the instrumental goal-directed avoidance conditioning reinforced the patients with AN to restrict food intake. Last, denying food could be a habit for patients with AN.

Despite these inferences sounding like they make sense; some comments could be considered further:

First, the determinant of cognitive factors seems to need to be considered even though memory and re-appraisal are information processing components. Therefore, the authors needed to explain how cognitive determinants will be treated in modern learning theory. 

We agree with reviewer #2 that further elaboration on this point would be helpful. Accordingly, a new section has been included, which is entitled: “3.1. How cognitive determinants are treated in modern learning theory” (pp.5-6):

“Cognitive factors influence learning and performance in complex ways. In the first case, to the extent that learning is cognitively reconceptualized in terms of mental representations that are created, assembled and/or altered to better reflect the external environment (Howard, 1999), AN may be characterized by the alteration of healthy representation. Here, an example is the overvaluation of eating, weight and/or shape, which are considered to be the core psychopathology underlying AN (APA, 2013). In the case of the assembly of abnormal mental representations, an example may be the food-related phenomenon of thought-shape fusion, specific and distinct cognitive distortions present in patients with eating disorders. It occurs when the thought about eating high-caloric food leads individuals to feel fatter (e.g., "Just thinking about eating a chocolate bar can make me gain weight") (Shafran, Teachman, Kerry & Rachman, 1999). An explanation advanced by modern learning theory is that thinking about these forbidden foods can activate the mental representation of sweet-fat foods, which excites the feared consequences of eating, including the internal body sensations, via a link with the catastrophic weight gain representation. Moreover, under this cognitive reconceptualization of learning, maladaptive automatic negative thoughts (e.g., ‘If I’m fat, people won’t like me’) are understood as a simple association between two mental representations, resulting in exaggerated or pathological responses (Baumeister & von Hippel, 2020). Likewise, it has been suggested that people can acquire associations by engaging in rule-based processing based on language and formal reasoning (De Houwer, Vandorpe & Beckers, 2005)

In the second case, cognitive factors also influence performance; for instance, in the control of food-related behaviours (Garcia-Burgos, 2022). Indeed, eating behaviour is often subject to sophisticated cognitive eating controls. One of the most widely practised forms of cognitive control over food intake is dieting, i.e., attempting to restrict intake as a means of weight regulation (Wardle, 1988). In AN patients, these cognitive regulations are especially important to overcome hunger sensations after long periods of deprivation. The problem is that anything that disrupts the cognitive control in people with a restricted diet (e.g., BP-AN) appears to unleash overeating (Polivy & Herman, 1985). Regarding the interplay between the cognitive content of learning and voluntary cognitive control processes in the context of food responses, both can be understood by a sequential pathway through a default-interventionist approach. Simpler automatic associative responses start, and then high-level processes are recruited when the simpler responses prove inadequate, particularly when conflict is detected (Garcia-Burgos, 2022). An example of conflict is when BP-AN patients refrain from their automatic tendency to eat attractive and pleasant chocolate in order to maintain incompatible goals in terms of weight status”.

Second, how does modern learning theory explain subclinical populations of AN? Please have more illustrations.

In order to further explain how modern learning theory accounts for subclinical populations of AN, the section “5.3. Multiple and different associations to explain anorexia nervosa subtypes “has been recast and a new figure has been added (Figure 4).

“In order to resolve the ambiguity and the behavioural approach-avoidance conflict, information provided by other cues (e.g., the context) is expected to be used (see Bouton, 1993, for a review). For instance, AN patients develop fear of becoming fat in situations predicting caloric eating such as the kitchen or mealtime. Then these contextual cues should further excite the fear association of sweet-fat cues with weight gain (Figure 4), promoting a high-arousal state that inhibits motivation to eat and food intake, as well as cognitive eating controls, to further maintain food rejection beyond physiological needs (Macht, 2008). However, in BP-AN, sweet-fat flavours are embedded not only in associations that excite the memory of the feared postingestive consequences of eating, but also in associations that serve to activate the memory of pleasant sensations of sweetness and hunger-reduction. It should be noted that preference for sweet food (which could also still be present pre-morbidly since food is rewarding for most people) is usually enhanced under food deprivation conditions. Then, the presence of interoceptive hunger cues will heighten the ability of sweet-fat flavours to retrieve the attractiveness of sweetness and overconsumption. In the absence of hunger cues, by contrast, this appetitive association should be reduced, making it more likely that these food-related cues will retrieve the memory of gaining weight, thereby reinstating food avoidance and food restriction. Thus, food restriction or binge eating will result in competition between the relative activation of feared and appetitive associations in each context”.

Third, it will be better to provide a framework of modern learning theory for AN in the article. The authors could have a table or figure to explain the mechanism of AN.

Accordingly, Table 2 has been added with the possible associative differences to explain subclinical populations of anorexia nervosa (AN), including the restricting type (R-AN) and binge-purging type (BP-AN), and healthy people.

Forth, please rewrite the abstract. In this abstract, I needed help understanding the purposes of the present study. Do learning theory and food restrict, which is the point of this review? Besides, what were possible explanations of the role of food restriction in AN based on learning theory? And how many articles were reviewed in the present study? Please revise this abstract and make it more concrete.

Both reviewers are in agreement with the idea of rewriting the abstract. This suggestion has been followed in the revised text, including the purpose, the point of this review and methodological aspects to make it more concrete.

“Improvements in clinical management of anorexia nervosa (AN) are urgently needed. To do so, the search for innovative approaches continues at laboratory and clinical levels to translate new findings into more effective treatments. In this sense, modern learning theory provides a unifying framework that connects concepts, methodologies and data from preclinical and clinical research to inspire novel interventions in the field of psychopathology in general, and of disordered eating in particular. Indeed, learning is thought to be a crucial factor in the development/regulation of normal and pathological eating behaviour. Thus, the present review not only tries to provide a comprehensive overview of modern learning research in the field of AN, but also follows a transdiagnostic perspective to offer testable explanations for the origin and maintenance of pathological food rejection. This narrative review was informed by a systematic search of research papers in the electronic databases PsycInfo, Scopus and Web of Science following PRISMA methodology. By considering the number and type of associations (Pavlovian, goal-directed or habitual) and the affective nature of conditioning processes (appetitive versus aversive), this approach can explain many features of AN, including why some patients restrict food intake to the point of life-threatening starvation and others restrict calorie intake to lose weight and binge on a regular basis. Nonetheless, it is striking how little impact modern learning theory has had on the current AN research agenda and practice”.

Round 2

Reviewer 2 Report

Thanks for the authors' revisions. I have no other comments.